# Cross-Cultural Validation of the Italian Version of the Bt-DUX: A Subjective Measure of Health-Related Quality of Life in Patients Who Underwent Surgery for Lower Extremity Malignant Bone Tumour

**DOI:** 10.3390/cancers12082015

**Published:** 2020-07-23

**Authors:** Mattia Morri, Peter Willem Bekkering, Marco Cotti, Matilde Meneghini, Enrico Venturini, Alessandra Longhi, Elisabetta Mariani, Cristiana Forni

**Affiliations:** 1Servizio di Assistenza Infermieristica, Tecnica e Riabilitativa, IRCCS Istituto Ortopedico Rizzoli, 40136 Bologna, Italy; marco.cotti@ior.it (M.C.); matilde.meneghini@ior.it (M.M.); enrico.venturini@ior.it (E.V.); cristiana.forni@ior.it (C.F.); 2Pediatric Physical Therapist & Postdoc researcher, Máxima sports & movement center, Princess Máxima Center for Pediatric Oncology, 3584 CS Utrecht, The Netherlands; w.p.bekkering@prinsesmaximacentrum.nl; 3Department of Chemotherapy, IRCCS Istituto Ortopedico Rizzoli, 40136 Bologna, Italy; alessandra.longhi@ior.it; 4Physical Medicine and Rehabilitation Unit, IRCCS Istituto Ortopedico Rizzoli, 40136 Bologna, Italy; elisabetta.mariani@ior.it

**Keywords:** osteosarcoma, quality of life, lower limb surgery, survey and questionnaire, validation study

## Abstract

The purpose of this study was to translate the English bone tumour DUX (Bt-DUX-Eng) questionnaire for lower extremity bone tumour patients, a disease-specific quality of life (QoL) instrument, into Italian and then examine the validity of the Italian version of Bt-DUX (Bt-DUX-It). The adaptation and translation process included forward translation, back-translation, and a review of the back-translation by an expert committee. The Bt-DUX-It was validated in a sample of adolescents treated for lower extremity osteosarcoma in Italy. Assessments included the Bt-DUX, the Toronto Extremity Salvage Score (TESS), and the European Organization for Research and Treatment Core Quality of Life Questionnaire of Cancer Patients (EORTC QLQ-C30). Fifty-one patients with a median age of 20 years (range: 15–25) completed the questionnaires. The mean Bt-DUX score was 70 (range: 16.30–100). The internal consistency of the overall score and that of the Bt-DUX-It was good: Cronbach’s α was 0.95. Spearman’s correlation coefficient between the Bt-DUX (total and domain scores) and EORTC QLQ C30 and TESS were overall moderate to good, reaching a *p*-value <0.01 in all cases. The Bt-DUX-It version is a useful tool for measuring QoL in patients with bone tumour and has similar internal consistency, construct validity, and discrimination as those of the Dutch and English versions.

## 1. Introduction

Malignant bone tumours like osteosarcoma and Ewing sarcoma mostly appear in the teenage years and in the long bones of the lower extremities. In Italy, the age-adjusted incidence of Ewing sarcoma and osteosarcoma in a population of 0–19 year olds is 4.3 for males and 2.6 for females per 1,000,000. The five-year survival rate is 60% [1]. Improved survival rates, surgical techniques, and the young age of patients affected by this disease have over time attracted growing interest in the functional outcome and quality of life (QoL) after surgery [2,3,4,5]. 

Measuring the quality of life in cancer patients is considered to be one of the main outcome parameters to measure the efficacy of surgery and chemotherapy treatment in addition to classical biomedical indicators [6,7,8]. The European Organization for Research and Treatment Core Quality of Life Questionnaire of Cancer Patients (EORTC QLQ-C30) [9] is one of the best-known tools to assess QoL in cancer patients. It has been translated into several languages including Italian and has been used in numerous studies for various types of tumour. To our knowledge, there is no specific assessment scale for bone tumour patients. Most studies on the outcome of surgery have so far mainly focused on basic daily activities or have used generic tools for QoL [7,10,11]. However, these tools fail to take into account a number of relevant issues, such as the patient’s own assessment of the cosmetic, functional, and emotional impact of the disease and its surgical treatment. These issues are included in a recently developed questionnaire DUX for bone tumours in children and adolescents (Bt-DUX) [12]. The Bt-DUX questionnaire was constructed as a disease-specific questionnaire, modelled on the generic DUX 25 QoL questionnaire (short version of the Dutch Children TNO-AZL Quality of Life Questionnaire/DUCATQOL). The Bt-DUX scores reflect patients’ personal impact: their individual values for cosmetic, social, emotional, and functional aspects of their life after surgery. The Dutch and English versions of the Bt-DUX [13] were found to be practically applicable with good internal consistency and validity and appeared to have added value regarding existing measures of quality of life in patients undergoing surgery for malignant bone tumours of the lower extremity in several studies [14,15,16]. To examine the validity of the Bt-DUX within bigger and/or international studies, it is important to translate it into other languages and validate it in other cultures/populations. Furthermore, in the Italian setting, it is important to have a specific tool for bone tumour patients that can be used in clinical practice.

The purpose of this study was to translate the English Bt-DUX (Bt-DUX-Eng) questionnaire into Italian and then examine the validity of the Italian version of the Bt-DUX (Bt-DUX-It).

## 2. Method

### 2.1. Translation, Adaptation, and Validation

To optimize the Bt-DUX questionnaire for use in an Italian setting, we translated and adapted it as far as possible according to internationally accepted and published guidelines for the process of cross-cultural adaptation of self-reported measures and recommendations by the World Health Organization (WHO) [17,18,19].

The process of cross-cultural adaptation aims to produce equivalency of content between the source language (in this project, English) and the target language (Italian), rather than simply linguistic/literal equivalence in preparing the new translated version of the scale. The process was carried out systematically and involved several stages, which are reported in sequential order. Translating the Bt-DUX into Italian took place at the Princess Máxima Center for Pediatric Oncology in the Netherlands. 

Stage I: initial translation. The first stage in the adaptation was the forward translation. The Italian forward translation of Bt-DUX was produced from the original English Bt-DUX by a bilingual translator whose native language was Italian and whose first foreign language was English. Instructions were given to the translator on how to approach translating, emphasizing conceptual rather than literal translations, as well as the need to use natural and acceptable language for the broadest audience. The translator, who had a background in research, had no knowledge about the concepts of bone tumour patients or quality of life and was considered to be a naive translator. The translator produced a written report of the translation with comments to highlight challenging phrases or uncertainties along with the rationale for final choices. The Italian version was discussed in a consensus meeting between the translator and the principal investigator of the study to reach a final Italian version (Eng-It/forward).

Stage II: back-translation. Working from the translated version of the questionnaire, and blinded to the original version, the Bt-DUX questionnaire was then translated back into the original English language by a bilingual translator whose native language was Dutch and who had extensive knowledge of English and Italian. The back-translator was neither aware of, nor informed of the concepts explored. The translator also gave a written report of the translation (It-Eng/backward) with comments included to highlight challenging phrases or uncertainties.

Stage III: expert committee. During a consensus meeting, the forward and backward translators and the principal investigator reviewed all the versions and comments, and after discussion, they reached consensus on the final wording and formatting to be used, resulting in the final harmonized Italian version of the Bt-DUX that was used for the validation process.

The readability of the Italian and English versions was measured with the Gulpease Index [20] and the Flesch–Kincaid Index [21], respectively. The Gulpease Index is a measure calibrated on the Italian language, ranging from 0 to 100, and a lower score indicates lower readability of the text. The Flesch–Kincaid Index is a readability test designed to indicate how difficult a passage in English is to understand, and a lower score indicates higher readability of the text.

Stage IV: test of the adapted version. The Italian version of Bt-DUX was validated among patients of the Istituto Ortopedico Rizzoli, Bologna, Italy. This part of the study had a cross-sectional design, requiring patient completion of the translated version of Bt-DUX and the three other questionnaires at the same time. All subjects gave their informed consent for inclusion before they participated in the study. The study was conducted in accordance with the Declaration of Helsinki, and the protocol was approved by the Ethics Committee of Rizzoli (N. 0008060, 03/07/2019). The study was registered on the ClinicalTrials.gov registry (N. NCT04074291).

### 2.2. Patients

All patients who underwent surgical intervention due to a malignant bone tumour in the leg were identified through hospital records and during scheduled follow-ups at the same hospital. Inclusion criteria were as follows: age between 15 and 25 years at the time of selection, time since surgery between 12 and 60 months, malignant bone tumour located in the leg, and limb sparing or ablative surgery. Patients were excluded if other medical conditions restricted their physical activities.

### 2.3. Assessment

In addition to the Bt-DUX-It [12], all patients received the EORTC QLQ-C30 [9] and Toronto Extremity Salvage Score (TESS) [22]. The disease-specific Bt-DUX concerns the patient’s subjective feeling about a specific aspect, using abstract faces (smileys) as answer categories. The expressions from very happy to sad (score 1–5) form a five-point Likert scale. The Bt-DUX consists of 20 questions, which cover social, emotional, cosmetic, and physical functioning domains. Single item scores are recoded and computed into raw total and domain scores. These raw scores are converted into total and domain scores, ranging from 0–100, with the highest scores indicating better QoL.

EORTC QLQ C30: The European Organization for Research and Treatment of Cancer (EORTC) Core Quality of Life Questionnaire is a well-validated instrument that assesses health-related quality of life (HRQOL) in cancer patients. It is used in clinical cancer trials in Europe, Canada, and the United States and has demonstrated high reliability and validity in different groups of cancer patients. The 30-item EORTC QLQ-C30 questionnaire is composed of scales that evaluate physical functioning and role functioning, as well as emotional, social, and cognitive functioning and global QoL. Three symptom scales measure fatigue, pain, and emesis, while six single items assess financial impact and physical symptoms such as dyspnoea, sleep disturbance, appetite, diarrhoea, and constipation. The time frame is the past week. The questions are formatted with either yes or no answers, or by using four-answer categories that range from 1, not at all, to 4, very much. The two questions on general health and global QoL are answered on a numbered visual-analogue scale from 1 to 7. The general health and global QoL ranged from 2 to 14.

The Toronto Extremity Salvage Score (TESS) is a validated and reliable disease-specific measure developed to evaluate physical disability in patients treated for extremity sarcoma. This self-administered questionnaire includes 30 items on activity limitations in daily life, such as restrictions in body movement, mobility, self-care, and the performance of daily tasks and routines. The degree of physical disability is rated from 1 (not possible) to 5 (without any problem). The raw score is converted to a score ranging from 0 to 100 points, with higher scores indicating no functional limitations.

### 2.4. Statistical Analysis

Based on the small sample size, descriptive statistics with the median score and range were used for the patients’ clinical demographic and outcome measures. Bar statistics were viewed to evaluate the distribution of the total and domain scores. Internal consistency of the Bt-DUX-It was determined by calculating Cronbach’s α and by computing the correlation between the four domain scores and the total Bt-DUX-It score (domain-total correlation). A Cronbach’s α value of 0.85 was considered as good [23]. In order to evaluate the preconceived domain structure of the Bt-DUX, an analysis of the item-domain correlation and Cronbach’s α of the four domains was performed. The construct validity of the Bt-DUX-It was determined by calculating the Spearman correlation coefficients between the Bt-DUX-It and the quality of life measures (EORTC QLQ-C30) and the functional ability domain (TESS). The relationship was most probably significant if the rho was closer to +1.00 or −1.00. The strength of correlation was interpreted according to “Guilford’s Rule of Thumb”. It states that a correlation coefficient that is less than 0.20 has a very low relationship. A correlation coefficient that falls in between 0.20 and 0.40 shows a low relationship, whereas a correlation coefficient that falls in between 0.41 and 0.70 shows a moderate relationship. A correlation coefficient that falls in between 0.71 and 0.90 shows a high relationship, whereas a correlation coefficient that has a value of more than 0.90 shows a very high strength relationship. Significance was set at *p* < 0.05.

Discriminate validity was evaluated by the ability of the Bt-DUX-It to discriminate between patients with worse and better functional status than the median value of the outcome measure. 

### 2.5. Sample Size

Musculoskeletal tumours are a rare disease. Therefore, for the translation and validation of TESS, a specific scale for this disease, fifty-three patients were enrolled in the study by Akiyama et al. [24] for the Japanese validation and 48 patients in the study by Saraiva et al. [25] (48 patients enrolled) for the Portuguese validation. For the present study, we thought a sample of 50 patients was sufficiently representative for this disease.

## 3. Result

### 3.1. The Translation Process

Due to the short and simple sentences in the original Bt-DUX, the translators encountered few difficulties. Agreement was generally reached with little discussion, and just a few phrases required more extensive discussion. In some cases, phrases that were closer to what Italian patients might say in their language were preferred over an exact word-for-word translation. Difficult phrases were for instance: about my life now, translated into “mia vita ora”; my sports capacities, translated into “mie capacitá nello sport”, and physical health/fitness, translated into “forma fisica”. Furthermore, some sentences needed to be carefully reworded to use the correct male/female form. 

The readability scores of the Italian and English versions were good, and the scales were easy to read. The readability did not significantly change after the translation process. According to the Gulpease Index score of 80, the Italian version was easy to read for children in their last years of primary school; according to the Flesch–Kincaid Index score, the English version was easy to read for third-grade elementary school children.

### 3.2. Patient Characteristics

The digital system of the hospital where the study was carried out provided the details of 132 patients who were eligible for the study. During their scheduled clinical follow-ups at the same hospital between July and December 2019, it was possible to contact 56 of them. Five patients were excluded due to severe complications that restricted their movement. A total of 51 patients were enrolled for the statistical analysis. Figure 1 summarises the enrolment process. The median age of the population was 20 years old (range 15–25), and the median follow-up from surgery was 38 months (range 14–60). Diagnosis was osteosarcoma in 54% of patients, and limb salvage surgery was performed in 93.2% of cases. Basal characteristics of the sample are shown in Table 1. The median total score of the Bt-DUX-It was 70 (range 16.3–100), for the TESS was 89.2 (range 38–100), and for the EORTC QLQ-C30 concerning the overall health was 83.3 (33.3–100). The specific score for each scale and relevant subgroups and items is shown in Table 2. The Bt-DUX-It did not show signs of ceiling or floor effects. Only two patients (3.9%) reached the top score, and none of them had the minimum score. The analysis of separate subgroups (emotional, social, cosmetic, and physical) did not show a ceiling or floor effect either (Figure 2). The physical subgroup was the one with the lowest score and showed a median of 55.

### 3.3. Internal Consistency

The internal consistency of the overall score and that of the Bt-DUX-It was good: Cronbach’s α was 0.95. The correlation of the overall scale score with separate domains was good: Spearman’s correlation coefficient ranged from 0.86 to 0.91 (*p* < 0.01). The correlation between various subgroups was moderate to good: Spearman’s correlation coefficient ranged from 0.61 to 0.84 (*p* < 0.01) (Table 3).

### 3.4. Construct Validity

The construct validity between total Bt-DUX with EORTC QLQ C30 in its three domains and TESS showed a Spearman’s correlation coefficient ranging from moderate to good (r = 0.56–0.76 with *p* < 0.01). The correlation of separate subgroups with the EORTC QLQ C30 and TESS showed a similar trend: Spearman’s coefficient ranged from moderate to good, reaching a significance of less than 0.01 in all cases. The Bt-DUX subgroup concerning functional skills showed a high correlation with respect to the TESS score (r = 0.79) and a moderate correlation with respect to the EORTC QLQ function (r = 0.69). In the assessment with the EORTC QLQ symptom scale, with regards to cosmetic and functional domains, the Bt-DUX showed a slightly lower correlation, 0.47 and −0.49, respectively, despite both being statistically significant anyway (*p* < 0.01).

### 3.5. Discriminant Validity

The sample was subdivided into two groups based on the median TESS scale, which was 89. An analysis of the total Bt-DUX scores and those of the subgroups for the two categories of patients showed how patients with a higher TESS score also had a higher BT-DUX score, which was statistically significant (*p* < 0.05). This trend was found not only for the total BT-DUX, but also for each subgroup (Table 4).

## 4. Discussion

The Italian version of the Bt-DUX to assess QoL and functional ability showed internal consistency, construct validity, and a good discrimination with results comparable to the original ones in the Dutch [12] and English [13] versions. Using this scale in clinical practice was simple. Patients were able to fill in the questionnaire in a short time, and the questions were easily understood, so there were no problems filling in the form. None of the questions went unanswered. 

The TESS score (median 89) was in line with that reported in the literature on comparable populations [26,27,28] with similar follow-ups. 

Cronbach’s α for the Italian version was 0.95, which was similar to that of the English and Dutch versions with respective values of 0.95 and 0.92. Spearman’s rho showed a good level of correlation in the internal validity among the four domains of the scale. Spearman’s correlation between the Bt-DUX and EORTC and TESS showed a range, in absolute values, between 0.56 and 0.76, which was moderate to good. In particular, in the Italian version, the scale showed a greater correlation with the TESS (r = 0.76 with *p* < 0.01) compared with the correlation of the English version (r = 0.49 with *p* non-significant) and Dutch version (r = 0.49 with *p* < 0.01). The Bt-DUX and TESS are exclusive scales for patients affected by bone tumours, and the correlation increased when taking into consideration the functional domain item of the Bt-DUX compared with the TESS, showing a correlation coefficient of 0.79.

The basic characteristics of the sample showed some differences with respect to samples tested in other studies concerning the validation of the Bt-DUX. In the present study, the percentage of male patients was 72%, which was greater than that of the Dutch version (49%) and that of the English version (59%). In 41% of patients of the present study, the type of operation performed was megaprosthesis compared with only 14% in the Dutch study. 

In the Dutch study, fifty-five percent of the patients enrolled underwent amputation compared with only 8% in the present study. A significant difference was the type of tumour: in the present study, the rate of Ewing’s sarcoma was 35.3% compared with 19% of the Dutch population and 12% of the English population. These differences among the studied populations might also explain some of the variations in the scores of the Bt-DUX and TESS questionnaires used in the three groups. At the same time, the high outcome of the scores showed how the scale can be used in different countries with different health contexts, thus favouring a general use of the scale. 

The correlation of the Bt-DUX with a commonly used QoL assessment scale in cancer patients, such as the EORTC, but not specific for bone-tumour patients showed results ranging from moderate to good. The EORTC has three main scores, one concerning general health status, one about functional ability, and one regarding the symptoms perceived by the patient. The lowest correlation with the Bt-DUX was the symptom domain, which ranged in absolute terms from a rho of 0.47 to 0.56. In many cases the follow-up time was quite long after the end of chemotherapy. Some items, such as nausea, lack of appetite, and tiredness, which are assessed in the symptom domain of the EORTC, probably over time had a reduced impact on the QoL of patients compared to those assessed by the Bt-DUX.

It is reasonable to suppose that the Italian version of the Bt-DUX is able to assess specific items of the QoL in bone tumour patients, such as the cosmetic aspect or the ability to perform sports activities, which might not be revealed by other tools. Sports and cosmetic aspects after major surgical intervention, in the long term, can considerably impact the patient’s QoL and should be taken into consideration in the scope of bone tumours. The EORTC symptom domain seems better able to assess disorders that appear in the initial phases of chemotherapy. Therefore, in bone tumour patients, the Bt-DUX might be a useful tool to be administered together with QoL scales already in use. 

The discriminating ability of the Bt-DUX was assessed by forming two study groups based on the median score of the TESS that resulted from the patients enrolled in the study. The group of patients with less functional recovery, TESS <89, was also the group with a lower Bt-DUX score, both as a total score (63.8 vs. 83.8) and single-domain scores of the Bt-DUX.

One limitation of the study was the low sample number and the possible selection bias. The rarity of the disease makes it difficult to enroll a large sample, so to reduce the risk of bias, the entire population available at the time of the study was enrolled consecutively. Furthermore, the study was monocentric, which, on the one hand, makes it harder to generalize the results, but, on the other, provides a more homogeneous sample with regards to surgical treatment and chemotherapy. 

## 5. Conclusion

The Bt-DUX is shown to be a useful tool for measuring QoL in patients with bone tumour, which is easy to use in clinical practice. The Italian version has similar internal consistency, construct validity, and discrimination as those of the Dutch and English versions. The availability of this scale in different languages is important to facilitate comparison between international and multicentre studies, which is fundamental in addressing a rare disease such as bone tumour. 

## Figures and Tables

**Figure 1 cancers-12-02015-f001:**
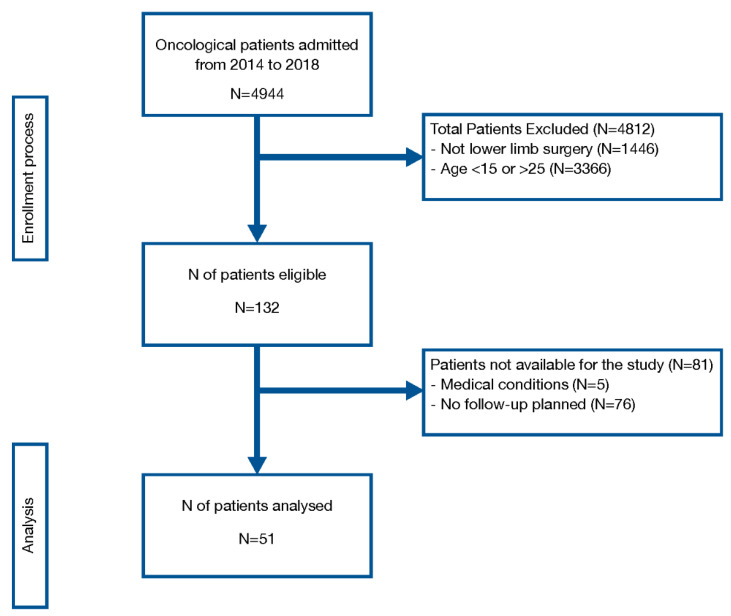
Enrolment process.

**Figure 2 cancers-12-02015-f002:**
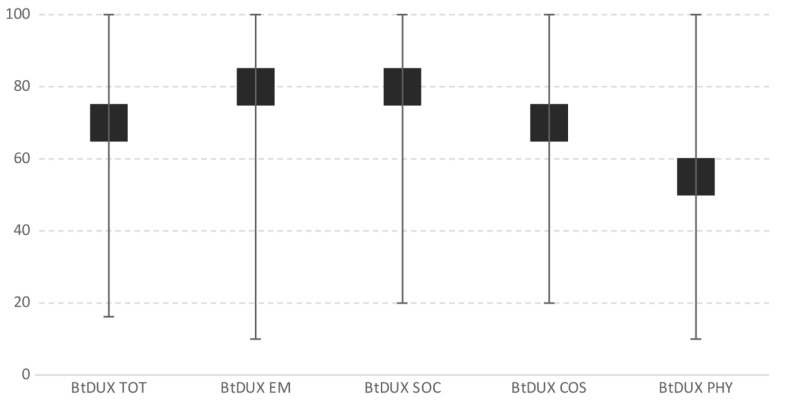
Ceiling or floor effects.

**Table 1 cancers-12-02015-t001:** Patient characteristics.

Characteristics	*N* = 51
Age, years, median (min-max)	20 (15–25)
Follow-up, months, median (min-max)	38 (14–60)
Sex, male (%)	37 (72.5)
Diagnosis	
Osteosarcoma, *n* (%) Ewing sarcoma, *n* (%) Other, *n* (%)	28 (54.9)
	18 (35.3)
	5 (9.8)
Surgery	
Prosthesis, *n* (%)	21 (41.2)
Bone graft, *n* (%)	23 (45.1)
Amputation, *n* (%)	4 (7.8)
Resection, *n* (%)	3 (5.9)
Site of the tumour	
Pelvis, *n* (%) Femur, *n* (%) Tibia, *n* (%) Foot, *n* (%)	3 (5.9)
	29 (56.9)
	17 (33.3)
	2 (3.9)

**Table 2 cancers-12-02015-t002:** Outcome measures. EROTC, the European Organization for Research and Treatment Core; TESS, Toronto Extremity Salvage Score.

Outcome Measure	Median (min–max) (0–100)
**EORTC**	
QoL, General Health QoL, Function QoL, Symptom	83.3 (33.3–100)
	88.9 (48.9–100)
	7.7 (0–38.5)
**Bt-DUX-It**	
Total score Emotional Social Cosmetic Physical	70 (16.25–100)
	80 (10–100)
	80 (20–100)
	70 (20–100)
	55 (10–100)
**TESS**	89.2 (38–100)

**Table 3 cancers-12-02015-t003:** Spearman correlation coefficients between the total Bt-DUX-It, Bone tumour Dux Italian version, scores and EORTC and functional ability. QLQ, Quality of Life Questionnaire.

Bt-DUX-It	EORTC QLQ General Health	EORTC QLQ Function	EORTC QLQ Symptom	TESS
Total score	0.68 **	0.70 **	−0.56 **	0.76 **
Emotional	0.66 **	0.63 **	−0.52 **	0.60 **
Social	0.70 **	0.63 **	−0.55 **	0.55 **
Cosmetic	0.59 **	0.60 **	−0.47 **	0.64 **
Physical	0.54 **	0.69 **	−0.49 **	0.79 **

** Correlation is significant at the 0.01 level.

**Table 4 cancers-12-02015-t004:** Bt-DUX total and domain scores in groups of patients with worse and better functional (TESS) status.

Bt-DUX-It	TESS < 89 *N* = 25	TESS > 89 *N* = 26	*p*-Value
Total score	63.8 (16.2–90)	83.8 (62.5–100)	<0.001
Emotional	65 (10–100)	90 (60–100)	0.001
Social	75 (20–100)	85 (55–100)	0.005
Cosmetic	60 (20–100)	85 (55–100)	<0.001
Physical	40 (10–75)	77.5 (35–100)	<0.001

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
