# Peer review of "Cross-Cultural Validation of the Italian Version of the Bt-DUX: A Subjective Measure of Health-Related Quality of Life in Patients Who Underwent Surgery for Lower Extremity Malignant Bone Tumour"

_cancers, 2020, doi:10.3390/cancers12082015_

Round 1

Reviewer 1 Report

In this manuscript the authors describe a cross cultural validation of the Italian version of the Bt-DUX. The Bt-DUX is a questionnaire that was developed to assess the patients' individual values of their life after surgery of a malignant bone tumour of the lower extremity. In particular, it focusses on four domains – the cosmetic, social, emotional, and functional outcome.

General comments

Primary malignant bone tumours are rare. Most of them like osteosarcoma or Ewing sarcoma are known to predominantly occur in adolecents. The adequate surgical treatment of malignant bone tumours has always been a challenging task for orthopedic surgeons. For the patients affected, it is of utmost importance to not only achieve a best possible clinical and functional outcome but also to guarantee a quality of life at a high level. In order to assess quality of life in these patients, several tools have been reported. However, these tools have thus far failed to fully assess all relevant issues such as the patients’own assessment of the cosmetic, functional and emotional impact of the disease and its surgical treatment. For this reason, the Bt-DUX has recently been developed that includes all relevant items in an easy to use questionnaire. The Bt-DUX however is so far only available in English and Dutch. For this reason, the aim of the authors was to translate the English Bt-DUX into Italian and examine the validity of the italian version.

As there is currently no comparably tool in Italian available, this is of high clinical relevance.

In general, this manuscript is written well and comprises a good structure. The introduction focusses on all relevant aspects, the patients and methods are comprehensible and the discussion evaluates adequately data from the translated questionnaire compared to other tools/scores.

Specific comments

Line 178. The authors report that the median age was 20 years ranging from 15 to 25. However, data provided in table 1 illustrates a range from 14 to 25 years. This needs to be corrected.

In summary, this is an interesting and nice work. I believe that the manuscript is likely to be of interest to the vision of clinicians, scientist and researches for this journal. I therefore recommend this manuscript for publication in this journal.  

Author Response

Dear Reviewer,

Thanks for your review. We have corrected the manuscript according to your comments.

Reviewer 2 Report

In the manuscript by Morri et al., titled "Cross cultural validation of the Italian version of the Bt-DUX: a subjective measure of health-related quality of life in patients who underwent surgery for lower extremity malignant bone tumour," the authors present their research findings regarding a QoL questionnaire (Bt-DUX) translated to Italian for the use of assessing patients with bone tumors. The background and introduction as to why this clinical tool is needed was well described. The methods section described the translation process in detail and the results were straight forward and very easy to understand. The discussion section was also well written pointing out the relevant caveats. 

I only have two minor criticisms: 

1) The authors do not discuss or compare the reading level of the English version and the Italian version. It would be of interest if it changed during the translation process (i.e. went from 10th grade reading level to 12th grade reading level). 

2) There were a handful of mispellings (i.e. "og" should be "of", "enrol" should be "enroll")

Other than that, very well done. 

Author Response

Dear reviewer,

thanks for your review. We have corrected the manuscript according to your comments.
